# The Influence of Mechanical Bowel Preparation on Volatile Organic Compounds for the Detection of Gastrointestinal Disease—A Systematic Review

**DOI:** 10.3390/s23031377

**Published:** 2023-01-26

**Authors:** Ashwin Krishnamoorthy, Subashini Chandrapalan, Sofie Bosch, Ayman Bannaga, Nanne K.H. De Boer, Tim G.J. De Meij, Marcis Leja, George B. Hanna, Nicoletta De Vietro, Donato Altomare, Ramesh P. Arasaradnam

**Affiliations:** 1Department of Gastroenterology, University Hospital Coventry and Warwickshire, Coventry CV2 2DX, UK; 2Department of Gastroenterology and Hepatology, Amsterdam Gastroenterology Endocrinology Metabolism, Amsterdam University Medical Centre, Vrije Universiteit Amsterdam, 1081 Amsterdam, The Netherlands; 3Department of Pediatric Gastroenterology, Emma’s Children Hospital, Amsterdam UMC, 1105 Amsterdam, The Netherlands; 4Institute of Clinical and Preventative Medicine, University of Latvia, LV-1586 Riga, Latvia; 5Department of Surgery and Cancer, Imperial College London, London SW7 2AZ, UK; 6Department of Chemistry, University of Bari Aldo Moro, 70121 Bari, Italy; 7Department of Surgery, University of Bari Aldo Moro, 70121 Bari, Italy

**Keywords:** volatile organic compound, colorectal cancer, mechanical bowel preparation

## Abstract

(1) Background: Colorectal cancer is the second commonest cause of cancer deaths worldwide; recently, volatile organic compounds (VOCs) have been proposed as potential biomarkers of this disease. In this paper, we aim to identify and review the available literature on the influence of mechanical bowel preparation on VOC production and measurement. (2) Methods: A systematic search for studies was carried out for articles relevant to mechanical bowel preparation and its effects on volatile organic compounds. A total of 4 of 1349 papers initially derived from the search were selected. (3) Results: Two studies with a total of 134 patients found no difference in measured breath VOC profiles after bowel preparation; one other study found an increase in breath acetone in 61 patients after bowel preparation, but no other compounds were affected. Finally, the last study showed the alteration of urinary VOC profiles. (4) Conclusions: There is limited data on the effect of bowel preparation on VOC production in the body. As further studies of VOCs are conducted in patients with symptoms of gastrointestinal disease, the quantification of the effect of bowel preparation on their abundance is required.

## 1. Introduction

Colorectal cancer (CRC) is the third commonest cancer and the second most common cause of cancer deaths worldwide [1]. Early diagnosis and treatment are associated with improved survival [2].

The current gold standard for the diagnosis of colorectal cancer and other lower gastrointestinal diseases is colonoscopy with biopsy and histological assessment. Colonoscopy is an invasive procedure with risks of discomfort, bleeding, and perforation. It is time-consuming, associated with negative patient experiences, and is a costly resource under high demand. Recent advances in CT technology have led to increased sensitivity and specificity in CT colonography to a level almost comparable to colonoscopy for lesions greater than 1 cm at least—but it is also time-consuming, expensive, involve exposure to ionizing radiation, and also requires air insufflation which can be uncomfortable [3]. Demand for colonoscopy services is set to rise even more in the coming years and will be coupled with an increased drive for the early detection of lesions and the associated increased survival. However, health resources are limited, and hence, the need for a non-invasive biomarker that can discriminate significant gastrointestinal disease from healthy subjects has never been greater [4].

### 1.1. Non-Invasive Biomarkers for Colorectal Cancer and Other Gastrointestinal Diseases

The ideal biomarker for colorectal cancer diagnosis should have the same desired features as all good screening tests. It should have a high sensitivity and specificity, it should be reliable and reproducible, and it should be acceptable to patients. In practice, this often equates to minimal discomfort and easy administration. Ideally, it should also be relatively inexpensive and give a quick result [5].

Using samples from patients such as blood, urine, faeces, or breath would be the easiest, minimally invasive way to achieve these criteria. Despite numerous scientific and technological advances, we still do not have the ideal test at this point in time—however, we will briefly review some of the proposed candidates:

### 1.2. Faecal Occult Blood Testing (FOBT) and Faecal Immunohistochemical Testing (FIT)

The earliest approach to CRC screening involved the use of a guaiac-based test card in order to look for occult blood in the faeces of patients. This method originated in the 1960s and relied on a certain amount of haemoglobin being present in faecal samples resulting in low clinical and analytical sensitivity, as well as susceptibility to influence by the ingestion of certain foods and drugs. These factors limited the popularity of faecal occult blood testing among some health providers. Nevertheless, it was still used as a screening tool for decades [6].

Since 2014, FOBT has been slowly but gradually superseded by faecal immunohistochemical testing (FIT) [7]. Numerous studies have confirmed increased diagnostic accuracy with FIT over FOBT—with improved clinical and analytical sensitivity, an increased ability to detect high-risk polyps such as advanced adenomas, and potential customization of the positivity threshold [8].

FIT tests are immunoassays that rely on an antigen–antibody complex forming between the test and the globin moiety of the human haemoglobin molecule. There are two types—qualitative FIT and quantitative FIT.

Qualitative FIT is most often based on a lateral flow immunochromatographic analysis test kit which gives either a “positive” or “negative” result—as with other point-of-care tests such as urine pregnancy tests or lateral flow test kits—for the detection of COVID-19.

Quantitative FIT, however, gives a numerical result by measuring the actual concentration of faecal haemoglobin. This allows for the potential risk stratification of patients with higher “positive” values than others.

At the test’s reported limit of detection—ranging from ≥2 μg Hb/g of faeces to ≥7 μg Hb/g—a recent meta-analysis of 11 studies in primary care providers reported a pooled sensitivity of FIT by 93.4% (95% CI 88.0% to 96.4%) and specificity of 76.9% (95% CI 67.7% to 84.0%) [9].

This illustrates why FIT has replaced FOBT and why it is becoming more popular in the use of screening and prioritizing patients with symptoms of the gastrointestinal disease for investigation [9,10].

### 1.3. Carcinoembryonic Antigen (CEA)

The carcinoembryonic antigen (CEA) is an intracellular adhesion molecule with a half-life of 1–3 days produced by epithelial tumour cells as part of angiogenesis. It is a glycoprotein discovered in the 1960s and can be detected in the blood of patients with adenocarcinoma of the colon [11].

Studies showed a low sensitivity and specificity of CEA for CRC, which, when combined with a low prevalence of CRC in the general population—led to unacceptably low positive predictive values, which have precluded its use in screening for CRC [12].

It has, however, been used with more success in the surveillance of patients with treated CRC to detect recurrent disease early [11].

### 1.4. DNA Methylation and ctDNA Methylation—mSEPT9

On a cellular level, colorectal cancer cells are found to have aberrant DNA methylation in more than 10% of the protein-coding genes when compared to normal colonocytes. This reflects the generalized hypomethylation, which leads to genomic instability, which is seen in cancer [13].

The septin-9 gene acts as a tumour suppressor gene. The hypermethylation of the CpG island in the promoter region of this gene inhibits the normal function of this gene leading to a consequent loss of the tumour suppressor function [13,14].

A recent meta-analysis looked at the diagnostic accuracy of mSEPT9 tests and calculated a 69% pooled sensitivity and 92% specificity for colorectal cancer. However, it showed poor diagnostic performance for precancerous lesions such as advanced adenomas and large polyps, a relatively high cost, and limited knowledge of the acceptability of the test to patients [14].

### 1.5. MicroRNAs (miRNAs)

MicroRNAs are small molecules of only 18–22 nucleotides in length that are involved in regulating gene expression through specific target messenger RNAs. There is evidence to support aberrant microRNA expression patterns in CRC with a theory that this plays a role in the pathogenesis of CRC development.

A recent systematic review found eleven studies evaluating the use of salivary miRNAs as biomarkers, with the most investigated disease being pancreatic cancer, but two studies looking at inflammatory bowel disease and colorectal cancer, respectively. There would appear to be still some way to progress in the development of salivary miRNAs, but if found to have good diagnostic accuracy, this would represent an easy and presumably acceptable test for patients as saliva is easier to obtain than other bodily fluids [15].

### 1.6. Volatile Organic Compounds

Volatile organic compounds (VOCs) are organic compounds that have a high vapour pressure or low boiling point at room temperature. This correlates with a high number of molecules in a gaseous form in the surrounding air, which can contribute to a particular odour or scent. Within the body, VOCs are abundantly produced as a result of distinct cellular metabolic pathways—and can be measured in the blood, urine, faeces, sweat, and breath [16].

There is a growing body of interest in their role as biomarkers for a variety of diseases. When considering their significance in gastrointestinal diseases, VOCs can arise from the body’s own cells but also from the vast array of micro-organisms within the gastrointestinal tract known as the gut microbiome and as a result of the complex interplay between both systems [17].

In certain gastrointestinal diseases, the VOC pattern changes and can distinguish these diseases from healthy subjects. VOC pattern changes reflect alterations in physiology and body metabolic processes. VOCs associated with certain cancers can be measured in faeces, urine, and exhaled breath, as well as in normal and cancer tissues [16,18].

Xiang et al. reviewed the literature and found a pooled sensitivity of 79% and specificity of 89% for VOCs in the diagnosis of colorectal and gastro-oesophageal cancers [19]. A case-control study by Altomare et al. in 2020 looked at the pattern of breath VOC in 83 colorectal cancer patients compared with 90 non-cancer control patients and identified fourteen VOCs (tetradecane, ethyl- benzene, methylbenzene, acetic acid, 5,9-undecadien- 2-one, 6,10-dimethyl (E), decane, benzaldehyde, benzoic acid, 1,3 bis(1-metiletenil) benzene, decanal, unidentified compound T22_75, dodecane, 2-ethyl-1-hexanol and ethanone, 1[4-(1-methylethenyl)phenyl]) with a 90% sensitivity and 93% specificity for identifying patients with colorectal cancer. This study showed that even at an early stage of the disease, the chemicals displayed good discriminatory ability with 86% sensitivity and 94% specificity [20].

In addition to this, Bosch et al. carried out a systematic review showing the high diagnostic value of VOCs in both colorectal cancers but also advanced adenomas, further favouring their role in early diagnosis [21]. Apart from intraluminal diseases of the gastrointestinal tract—more recent data have also emerged on the utility of VOCs in patients with hepatocellular carcinoma (HCC). Sukaram et al. carried out a study in 97 HCC patients compared with 111 controls and identified six VOCs (acetone, 1,4-pentadiene, methylene chloride, benzene, phenol, and allyl methyl sulfide) with a combined sensitivity and specificity of 76.5% and 82.7% [22]. Bannaga et al. reported urine VOCs that discriminated HCC patients from non-HCC patients (in a group with patients who had liver fibrosis) with a specificity of 95% and a sensitivity of 43% [23].

At this stage of research into VOCs as biomarkers, the exact metabolic pathways from which they originate have not yet been confirmed. However, we know that certain factors, such as diet, affect VOC production. For example, Rossi et al. [24] carried out a randomized controlled trial that showed a significant change in faecal VOCs in patients who followed a low FODMAP (Fermental Oligosaccharides, Disaccharides, Monosaccharides, and Polyols) when compared to a sham diet. Couch et al. [25] carried out a study that showed that subjects with a history of alcohol excess exhibited significant alterations in faecal VOCs as well when compared to healthy controls.

These findings implicate the microbiome as a potential origin for faecal VOC signatures.

The gut microbiome has been implicated in maintaining physiological health. For example, one of the functions of the microbiome is in the fermentation of non-digestible substrates such as dietary fibre and intestinal mucus resulting in short-chain fatty acids (SCFA) and gases [26]. Butyrate is a major source of energy for colonocytes, and proprionate has been shown to induce apoptosis in cancerous liver cells [27]. Differences in microbiome composition and function have been associated with many chronic diseases—including autoimmune diseases, cardiovascular disease, and gastrointestinal diseases—but we need a greater understanding of the role of the microbiome in these diseases.

VOCs may also arise from colonocytes or tumour cells themselves and be secreted into the bloodstream before being filtered by the kidneys and found in the urine.

The study of metabolomics in the gut and microbiome leads to the potential clinical application of VOCs. [28]

Colonoscopy is a finite resource under large demand in the United Kingdom, and so the use of faecal immunohistochemical testing (FIT) is presently being used as a triage tool for patients who are referred to with lower gastrointestinal symptoms [9].

However, the overall sensitivity and specificity of FIT lie at 90% and 87% respectively, and studies have reported false negative rates of up to 14%. There is also concern over its sensitivity in picking up high-risk polyps or early colorectal tumours, and as a result, we are unable to safely substitute colonoscopy completely with FIT testing. Data have shown that the combination of FIT and VOC testing can increase diagnostic accuracy and be a better screening tool—with the probability of having colorectal cancer after a negative FIT test and a negative VOC test of 0.1% [29]. In the future, further studies examining the use of FIT and VOC testing together as a combined screening tool may help to mitigate demand from overburdened services.

In order to obtain optimal data from studies going forward, a detailed knowledge of factors influencing VOC measurement is required. Some factors will be non-modifiable, such as genetic factors or the presence of obesity or diabetes mellitus. However, others can be controlled to some extent or account for factors such as diet, smoking, drug ingestion, and sampling conditions such as temperature [30,31].

### 1.7. VOC Detection Techniques

At this early stage in VOC research, there is variability in the detection of in vitro VOCs with four of the main techniques described below:Gas Chromatography-Mass Spectrometry (GC-MS)

Gas chromatography-mass spectrometry (GC-MS) is perhaps the most common technique encountered in the literature for the measurement of VOCs. This method can identify single VOCs and, therefore, give us information about the underlying metabolic pathways leading to these compounds. Potential disadvantages of this method include the cost and the need for highly trained operators as well as heavy and expensive equipment that cannot be used practically in a clinical setting [32].

Field Asymmetric Ion Mobility Spectrometry (FAIMS)

This is a refinement of mass spectrometry—by passing a voltage between plates, some ions drift and hit the plates, while others stay in between, allowing separation of a complex mixture of gases by their differences in mobility in high electric fields resulting in increased sensitivity and the detection of minute changes in VOC composition [33].

Selected Ion Flow Tube Mass Spectrometry (SIFT-MS)

This is an emerging form of a mass spectrometry technique that combines chemical ionisation reactions with mass spectrometric detection to rapidly quantify VOCs [34].

Gas-Chromatography coupled to Time-of-Flight Mass Spectrometry (GC-TOF-MS)

This technique involves the separation of a sample mixture by gas chromatography before accelerating ions with different masses to the same kinetic energy before measuring the time taken for each ion to reach a sensor at a known distance. Since this time will be dependent on the mass-to-charge ratio of the ion, this allows for the accurate quantification of compounds. Again, as with pure GC-MS, it is a time and labour-intensive process with the need for specialised equipment and well-trained staff [35].

Electronic—Nose/“E-Nose” Technology

This is perhaps one of the most exciting techniques with regard to the clinical application of VOCs as biomarkers.

In the last thirty years, there has been a lot of work looking into methods of artificial olfaction. Electronic noses, or “e-noses”, are devices that are designed to work similar to the mammalian nose by sensing mixtures of organic compounds and generating an output of data.

They typically consist of an array of sensors, along with an information processing unit: pattern-recognition software and reference library databases. A flow diagram of how they process data from sample materials is given in Figure 1.

The sensors create an output of data that goes to the information-processing unit or artificial neural network (ANN),—which uses pattern recognition algorithms that cross-reference against the reference library database—to then discriminate various aromas in the sample material. They have been used in the industry and manufacturing in order to achieve quality control, product consistency, gas leak detection, and effluent monitoring [36].

The sensors are made from nanomaterials that are developing at high speed. There has been recent work on nanosensors that can account for the high humidity of human breath samples while retaining good performance in the detection of VOCs, such as acetone, and this leads to the exciting possibility of high-performance wearable sensing devices in the future [37].

As VOC research progresses, so does research into the use of e-noses and their application in detecting diseases from human samples. It is widely recognised that we have not yet identified a single compound in the breath, urine, or faeces that discriminate colorectal cancer from other diseases, which could be likened to the “holy grail” of VOC research. Instead, there is an altered pattern of compounds that can be sensed as a different “aroma”, and this is where e-noses can play a significant part in the non-invasive diagnosis of disease.

They are not designed to readily identify individual compounds or reflect individual metabolic pathways, which may be displayed as a relative disadvantage [38,39].

However, this must be measured against the fact that they are incredibly simple to use and inexpensive compared to the analytical techniques detailed above. Staff can be easily trained to operate these devices, so this increases their wider applicability.

### 1.8. Mechanical Bowel Preparation and VOCs

The use of mechanical bowel preparation for the cleansing of the bowel and adequate visualization of the lumen during colonoscopy has long been used to try to ensure a high yield of disease diagnosis.

In 2016 the British Society of Gastroenterology (BSG), the Joint Advisory Group on GI Endoscopy (JAG), and the Association of Coloproctology of Great Britain and Ireland (ACPBGI) published joint guidelines on key performance indicators and quality assurance standards for anyone potentially carrying out diagnostic colonoscopy for patients [40].

This was an attempt to try and tackle the perceived variability in practice and improve quality through objective measures.

One of the key quality indicators they advocated was a minimal caecal intubation rate of 90%, with photo documentation of reaching the terminal ileum or caecal landmarks essentially denoting a complete colonoscopy.

Studies report that failure to reach the caecum in colonoscopy can be attributed to suboptimal bowel preparation in 20–32%. Furthermore, a study of 93,000 patients showed that the detection of small colonic polyps (<10 mm) was positively correlated with the quality of bowel preparation [41].

Hence, the use of adequate mechanical bowel preparation is of the utmost importance and is associated with increased rates of diagnosis for significant large bowel diseases, including adenomas and colorectal cancers.

### 1.9. Different Categories of Bowel Preparation

Bowel preparation can be divided into three categories—isosmotic, hypoosmotic, and hyperosmotic agents.

Iso-osmotic agents are generally based on polyethylene glycol (PEG). This compound works as an “osmotic laxative” by causing water retention in stools, thereby softening the stool and increasing the number of bowel movements. This class of bowel preparations includes both high-volume PEG and low-PEG preparations. High-volume PEG preparations have nonfermentable electrolyte solutions in order to avoid significant fluid and electrolyte shifts for the patient.

This is better for patients with conditions where fluid homeostasis is significantly perturbed, such as congestive cardiac failure, cirrhosis with ascites, or chronic kidney disease.

Hypo-osmotic agents are not often used.

Hyper-osmotic agents are most commonly based on sodium picosulphate. They work by pulling water from the body into the bowel lumen. Due to the potential for dehydration and electrolyte disturbances, such as hyponatraemia and hypokalaemia, they are being used with reduced frequency [42].

The European Society of Gastrointestinal Endoscopy (ESGE) does not recommend the routine use of hyper-osmotic agents such as sodium picosulphate, mainly due to the electrolyte disturbances described above. Instead, polyethylene glycol (PEG)-based regimens are recommended and are the most likely to be used in today’s practice [43,44].

What we can learn from the above is that patients who are enrolled in studies looking at FIT and VOCs as potential screening tools for colorectal disease are, therefore, likely to be taking mechanical bowel preparation at some point; the question of how bowel preparation affects VOC measurements assumes the utmost importance. Mechanical bowel preparation is certainly shown to reduce the abundance of gut bacteria in the colon, at least transiently [45]. Since the origin of VOCs may be, in part, from the gut microbiome [46], then potentially bowel preparation could artificially affect VOC measurement and abundance, which would have implications for their clinical application as potential disease biomarkers or screening tools. In this paper, we aim to review the available literature to see if mechanical bowel preparation has been demonstrated to influence VOC measurement.

## 2. Materials and Methods

### 2.1. Eligibility Criteria

We looked for original, peer-reviewed articles that were published in the English language in the Pubmed/MEDLINE, Embase, and Cochrane databases over the past twenty years, from December 2001 to December 2021. All studies should have reported on VOCs measured from human participants of any media, urine, stool, breath, or blood to merit inclusion. Studies should have carried out statistical analysis on the effect of mechanical bowel preparation on the VOC they measured.

There was no lower or upper limit on the number of patients that could be included in the study due to the relative paucity of reported data in the literature. For the same reasons, it was decided not to limit the patient population, and both healthy subjects and those with the confirmed disease could be included.

Papers that were not in English and conference abstracts were excluded. Full inclusion and exclusion criteria were as below:

#### 2.1.1. Inclusion Criteria

Studies comparing breath/urine/faecal VOC results in human patients before and after mechanical bowel preparation.Statistical analysis of the results and a conclusion drawn from this.

#### 2.1.2. Exclusion Criteria

VOC Samples solely taken before bowel preparation/colonoscopy.Papers not in English.No statistical analysis/inadequate numbers for conclusive results.Conference abstracts.

### 2.2. Information Sources

Initial informal searches were carried out in December 2021 on the Pubmed/MEDLINE database looking for articles related to VOC research in colorectal cancer. The reference lists of articles in the initial papers found and recurrent searches of Pubmed were used to inform the formal search strategy and search string which is detailed below.

### 2.3. Search Strategy

An electronic search of the literature was conducted on the Pubmed/MEDLINE, Embase, and Cochrane databases with the following search string:

The formal MESH search terms were as follows: ((((VOC) OR Volatile Organic Compounds [Abstract]) OR Volatile Organic Compounds [Title]) AND Bowel Preparation) OR Bowel Preparation [Abstract]. The abstracts of the resulting papers were screened for relevance to the research question and either included or excluded according to the criteria below by two independent authors.

### 2.4. Selection Process

The study selection process is summarised by the PRISMA diagram in Figure 2. A total of 301 duplicate records were removed before the screening process started. A total of 1048 articles were extracted from the three databases.

Two authors (AK and SC) screened the titles and then the abstracts sequentially. The screening was carried out independently, and clearly, ineligible studies were excluded due to incorrect study designs or endpoints to fulfill the criteria needed for this review.

Where both authors (AK and SC) agreed on the exclusion of the study—“congruent exclusion decisions”—this led to the straight exclusion of the study. Incongruent exclusion decisions led to an abstract examination and then a review of the full paper.

The papers were then read in full by either AK or SC, with AB acting as an independent mediator in the case of disagreement.

This left 128 full papers, which were then assessed against the inclusion/exclusion criteria, leaving only five studies for analysis.

## 3. Results

The search yielded five publications that fulfilled the inclusion criteria [47,48,49,50,51].

The overview of the structure of the five studies is given in Table 1, and the specific volatile compounds measured in each study are in Table 2.

Markar et al. [52] analysed VOCs using SIFT-MS in breath samples of 150 patients with 50 colorectal cancers, 50 patients with non-cancerous conditions of the gastrointestinal tract (positive controls), and 50 patients with a normal gastrointestinal tract (negative controls). This study showed that propanal appeared to distinguish patients with colorectal cancers from other subjects with a diagnostic accuracy ranging from 79 to 90%.

They looked at 112 patients who underwent bowel preparation and did not find any significant difference in propanal concentrations as a result of mechanical bowel preparation, using multivariate linear regression analysis.

The strengths of this study were in the discovery of a single VOC that was significantly altered in cancer patients (most other studies have only found differences in the pattern of VOCs) and the validation of their findings in two prospective cohorts of patients with colorectal cancer. One potential limitation is that there was an increased proportion of control patients that received bowel preparation. Additionally, this was a single-centre study with the use of the SIFT-MS analytical platform instead of the more common GC-MS; thus, the clinical and analytical aspects are relatively limited in this respect.

Francis et al. [47] carried out a feasibility study also looking at whether breath VOC analysis (SIFT-MS) could predict postoperative ileus after laparoscopic colorectal tumour resection. As with the first paper, a separate analysis looked into patients receiving enemas and mechanical bowel preparation and the measured abundance of VOCs. There was no significant difference in baseline VOC concentration between patients who had enemas, mechanical bowel preparation, or those that had no bowel preparation. There was also no difference in VOC concentration in patients who had oral intake prior to measurement versus those patients who were starved. The strengths of this paper were the statistically significant results reported with no change in VOC profile; the measurement of multiple VOCs were also used rather than a single VOC marker which is more applicable to the generalised measurement of VOCs. Again, the major limitations were that this was a single-centre study with relatively small numbers of patients studied (22 in the enema/mechanical bowel preparation group), and there was no comparison of the same patients before and after bowel preparation.

Arasaradnam et al. [48] carried out a study in 2012 on the changes in VOC profile in patients undergoing complete versus partial bowel cleansing using FAIMS and electronic nose technology. There were a total of 23 patients—of which 14 underwent complete mechanical bowel preparation, and 9 had partial bowel cleansing (by phosphate enema). Samples were compared prior to bowel cleansing and 48 h after the commencement of bowel cleansing, with a modulation in VOCs/gases seen with both FAIMS and e-nose analyses with the greatest effect at 48 h. There was a significant but partial recovery of abundance in VOCs which was not seen at 48 h, with urine samples taken at 2 weeks after bowel preparation, suggesting that this may only be a transient effect. The advantage of this study was in the use of the electronic nose measurement and in measuring VOCs in urine which reflects VOCs quickly absorbed into the bloodstream. The main limitation of the study is that all patients had a normal examination of the colon. No patients with colorectal cancer or other diseases of the gastrointestinal tract were included. No patients with colorectal cancer or other diseases of the gastrointestinal tract were included.

Leja et al. [49] carried out a study looking at the breath VOC profile in three patient groups, one of which was a group of 61 patients referred for colonoscopy. Patients with cancer or inflammatory bowel disease were not included, but seven patients with high-risk colonic polyps and seven with low-risk colonic polyps were included in the analysis. The VOC profile before and after polyethylene glycol-based bowel cleansing (median time of 5 days between samples) was compared with only one statistically significant difference: an increase in acetone. The authors suggested that this result may be due to ketosis in fasting prior to the colonoscopy rather than as a result of microbiome eradication. Since all other compounds were unaffected by bowel preparation, they concluded that bowel preparation did not have a significant effect on the microbiome and VOC production. As with the 2012 paper by Arasaradnam, no confirmed cancers were included in this group.

Finally, Woodfield et al. [50] carried out a multicentre prospective diagnostic accuracy study with the aim of developing a breath test to accurately diagnose colorectal cancer.

This was the most recent study found in our search.

In this study, 162 colorectal cancer patients were recruited alongside 1270 positive control subjects (with a variety of conditions including benign anorectal pathology such as haemorrhoids, inflammatory bowel disease, and low-risk colorectal polyps); there were a total of 1432 patients overall. Breath samples were collected from patients in thermal desorption tubes and sent for GC-MS analysis.

Twenty-five endogenous VOCs were identified from the study with a good predictive ability for CRC. From this initial pool, a diagnostic model comprising fourteen endogenous VOCs and body mass index was found to predict CRC with an area under the ROC curve of 0.91, with a sensitivity of 83%, specificity of 88%, and a negative predictive value of 96%.

Of the 162 patients with CRC in the study, 124 received mechanical bowel preparation, while 37 did not (with one patient unknown).

Both age and the use of mechanical bowel preparation were specifically declared as factors that were not predictive features or confounding factors in the VOC-based model.

The strengths of this study were the large sample size with patients recruited from seven different centres. Furthermore, the use of analysis to discriminate exogenous from endogenous VOCs—with the subsequent use of endogenous VOCs only in analysis—represents an attempt to remove some of the external factors that could confound their potential as biomarkers.

As with all the other studies, the role of mechanical bowel preparation was not a primary outcome of the study; therefore, little data was reported on this apart from what is summarised above. The type of bowel preparation also was not formally mentioned in the study.

Table 3 summarises the characteristics of each study as described above; with description of which patients were compared and whether a significant effect on VOC abundance was found after use of mechanical bowel preparation.

## 4. Discussion

The analysis of VOC patterns in disease states is an emerging field of research interest with the potential for clinical application; however, it is clear that there is a paucity of data in the literature related to the effect of bowel preparation on VOC measurements in the body.

This reflects the early stage that we are at in terms of research into VOCs as biomarkers for certain diseases.

The studies that we extracted from the literature did not give sufficient data on the VOC measurement for us to perform a thorough meta-analysis.

Heterogeneity was also a challenge for our research question when considering the four papers that we extracted from the literature.

The different papers that we analysed looked at different patient groups, some at healthy patients while others looked at colorectal cancer patients undergoing an operation or high-risk polyps. Four papers examined breath VOCs, while one focussed on urine VOCs. Finally, the type of bowel preparation was not specified in any apart from Leja et al. [49], who used a polyethylene glycol-based medication—which is the most common type in general use.

Nevertheless, there are some promising findings that have not been reported before to our knowledge. Our study suggests that bowel preparation does not significantly affect the breath VOC profile of either normal subjects or those with colorectal cancer. The urinary VOC profile was affected. This may suggest that VOCs of different media are affected differently by external factors. Breath VOCs may be more stable or resistant to the influence of external factors than urinary VOCs in the short term.

### The Exact Origin of VOCs

Another explanation lies within the potential origin of VOCs in the disease state of colorectal cancer. We know that bowel cleansing leads to reduced diversity and the abundance of certain gut bacteria. Studies also show that this is a transient effect and that these bacterial populations are likely to recover after 2–4 weeks [45]. However, if VOCs are arising from metabolic pathways in tumour cells or colonocytes rather than gut bacteria, this could account for the sparing of VOC profiles after bowel preparation. A study by Bosch et al. showed that 3 months after polypectomy for colorectal adenomas, the faecal VOC profile was “normalised”; patients with adenomas and controls were discriminated by VOC analysis prior to polypectomy, but after polypectomy, their faecal VOC profiles were similar. This study suggests the origin of biomarker VOCs to be the tumour rather than the microbiome, and this would fit with the findings in our study [51].

When looking at HCC development in the liver, Bannaga et al. hypothesised aberrant Cytochrome P450 (CYP450) metabolic pathways resulting in different volatile compound production and excretion in the urine [23]. In a preliminary study from De Vietro et al., the VOCs exhaled by colorectal cancer patients were analysed and compared with those produced by the cancer tissue and normal colonic mucosa biopsies harvested from surgical specimens using a headspace solid-phase microextraction (HS-SPME). Benzaldehyde, benzene ethyl, benzene methyl, butanoic acid, dodecanoic acid, indole, nonanal, octanoic acid, pentanoic acid, phenol, and tetradecane were the VOCs most frequently detected both in the exhaled breath and in those secreted by tissues. The cancer tissue and normal colonic mucosa from the same patient produced a similar VOC pattern but with different fingerprints. In fact, the concentrations of benzaldehyde, benzene ethyl, and indole were significantly different in cancer tissues with respect to the normal colonic mucosa. Interestingly, the quantity of indole, which is preferentially produced by the microbiota, was lower in normal mucosal and exhaled breath than in cancer tissue [18].

Liu et al. carried out a study in 2019 using solid-phase microextraction (SPME) and GC-MS analysis on the blood and tumour cells of mice with moderately advanced colorectal adenocarcinoma. Decanal, 2,4-dimethyl-heptane, and twelve other compounds were significantly increased in these samples; this led the authors to propose that these directly increased VOCs could be used as potential biomarkers for the disease [53].

With regard to the altered state of gut microbiota or “dysbiosis” that exists in patients with colorectal cancer, there is much that is yet to be learned. It is theorised that dysbiosis is a chronic state which occurs over many years and represents the depletion of certain bacterial species with the dominance of other species, which leads to a tendency toward tumorigenesis in the colorectum. Experimental evidence points towards specific bacterial strains associated with CRC, such as fusobacterium nucleatum, Escherichia coli, and Bacteroides fragilis [54].

While we need to know more—if the theory of tumorigenesis as a result of dysbiosis is true—this could be compatible with tumour cells as the eminent source of biomarker VOCs rather than the microbiota themselves.

These findings potentially support our theory that VOCs, correlating with disease states such as cancer, arise directly from tumour cells rather than as products of gut bacteria. Dysbiosis or imbalance in the microbiome still exists in the disease state, and this is not refuted; however, the metabolic pathways responsible for VOCs may be coming directly from tumour cells.

## 5. Conclusions

The knowledge of metabolic pathways in tumorigenesis and tumour growth—as well as the interaction between the gut microbiome and host cells—is still in its infancy. We need more studies to elucidate these pathways.

Furthermore, as we have alluded to in the introduction, detailed knowledge of the factors affecting VOC abundance and measurement is essential as more studies are carried out looking at their diagnostic accuracy as biomarkers for certain human diseases and conditions such as cancers.

As colorectal cancer is so common in the Western world and as it is not straightforward to diagnose as an early-stage disease—it would appear that VOC research in this field will only become more common in the next few years, and hence, it is imperative to know if external factors such as smoking, diet, drugs, or indeed the use of mechanical bowel preparation, unquestionably affect VOC measurement and may affect their validity as biomarkers.

In the past, studies have analysed VOCs in patients prior to taking bowel preparation with the assumption that it interferes with the value of the data obtained. Our study suggests this may not be the case.

Going forward, our specific research question could be answered by incorporating a specific analysis of breath, urine, or stool VOC profiles before and after bowel preparation in matched subjects, as was carried out in two of our five identified studies. This could be performed in both diseased subjects, healthy patients (negative controls), and positive control patients. Ideally, a polyethylene glycol-based preparation would be studied as this is the most common and clinically relevant type of preparation with minimal side effects to the patient when compared with others.

This would give more information on the true effect of bowel preparation on the VOC profile, and hence, clarify the ideal timing and conditions for sampling urine, stool, or breath in patients as VOCs develop as clinically relevant biomarkers for colorectal disease.

## Figures and Tables

**Figure 1 sensors-23-01377-f001:**
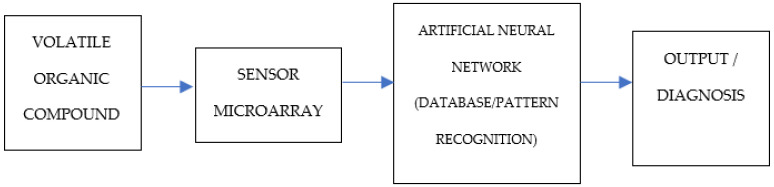
Schematic representation of an electronic nose.

**Figure 2 sensors-23-01377-f002:**
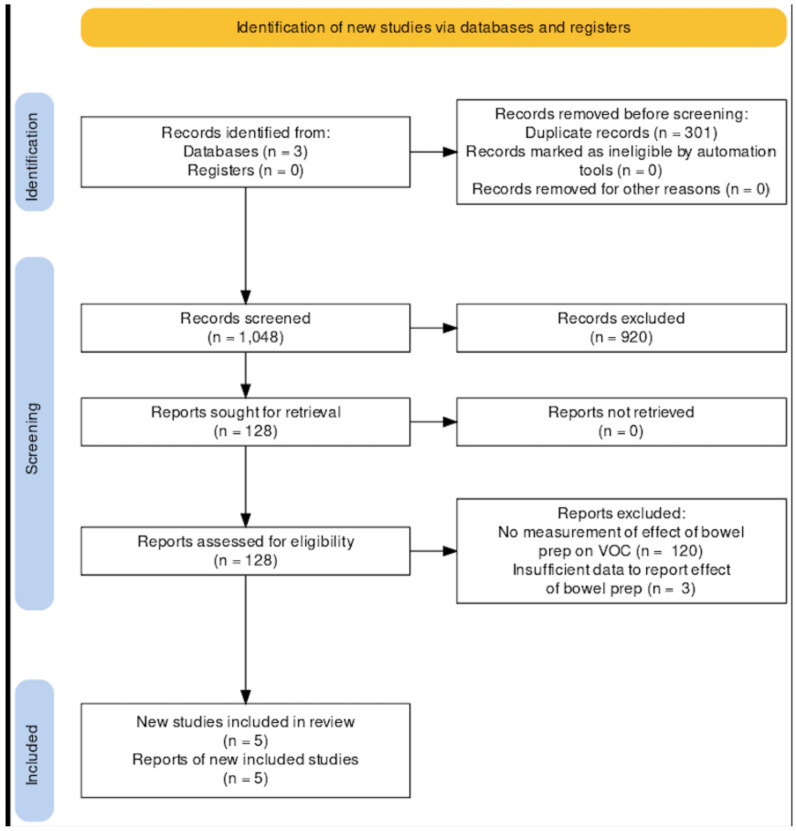
PRISMA flow diagram showing the search and selection process of the reviewed studies.

**Table 1 sensors-23-01377-t001:** Summary of VOC media, compounds, and effects of bowel prep on VOCs.

Study	VOC Medium	Measurement Method	Significant Effect of Bowel Prep on VOCs	Number of Patients Analysed
Arasaradnam et al., 2012 [48]	URINE	FAIMS	YES	23
Leja et al., 2016 [49]	BREATH	GC-MS	YES	61
Markar et al., 2019 [52]	BREATH	SIFT-MS	NO	112
Francis et al., 2019 [47]	BREATH	SIFT-MS	NO	22
Woodfield et al., 2022 [50]	BREATH	GC-MS	NO	161

**Table 2 sensors-23-01377-t002:** Specific chemicals measured in each study.

Study	Specific VOCs Measured in Study
Arasaradnam et al., 2012 [48]	Hydrogen sulphide (Urine)
Leja et al., 2016 [49]	Acetone (Breath)
Markar et al., 2019 [52]	Propanal (Breath)
Francis et al., 2019 [47]	Methane, Hydrogen, Dimethyl sulphide, Acetic acid, Ethanol, Methanol, Ammonia, Hydrogen sulphide (Breath)
Woodfield et al., 2022 [50]	Dimethyl sulphide, 1-Penten-3-ol, Heptane, cyclopropane, (Breath)

**Table 3 sensors-23-01377-t003:** Illustration of articles and effect on VOCs.

Study	Index Group vs. Control	Cases	Controls	Matched Patients?	Sample Medium	Significant Effect on VOCs?
Arasaradnam et al., 2012 [48]	Picolax + moviprep vs. after	19	19	YES	Urine	YES
Leja et al., 2016 [49]	Before polyethylene glycol vs. after	61	61	YES	Breath	NO
Marker et al., 2019 [52]	Mechanical vs. no bowel prep	30	82	NO	Breath	NO
Frances et al., 2020 [47]	Enema/mechanical vs. no bowel prep	22	25	NO	Breath	NO
Woodfield et al., 2022 [50]	Mechanical vs. no bowel prep	124	37	NO	Breath	NO

Column 5 (Matched patients): Green represents the study comparing effect of mechanical bowel preparation on the same patients (matched) and yellow represents different patients (non-matched). Column 6 (Sample medium)—Yellow represents urine as the VOC medium; Blue represents Breath as VOC medium. Column 7—Red represents a significant effect on VOC abundance seen in the study; and green represents no significant effect on VOCs seen as a result of mechanical bowel preparation in the study.

## Data Availability

Not applicable.

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
