# Peer review of "The Influence of Mechanical Bowel Preparation on Volatile Organic Compounds for the Detection of Gastrointestinal Disease—A Systematic Review"

_sensors, 2023, doi:10.3390/s23031377_

Round 1

Reviewer 1 Report

The authors have provided an excellent review and critique of currently available methods for VOCs analyses, with a good understanding of theirs capabilities and limitations. The importance of the measurements has been placed in context where the VOCs measurement was functioned as biomarkers for certain diseases.

The paper is well written and well presented with care. I recommend that it be published in sensor with no further modification.

Author Response

Thank you Sir/Madam

Reviewer 2 Report

In this paper, the authors identify and review the available literature on the effect of mechanical bowel preparation on VOCs production and measurement. Two studies with a total of 134 patients found no difference in the measured breath VOCs profile after bowel preparation; another study found an increase in breath acetone in 61 patients after bowel preparation, but no other compounds were affected. The last study showed an altered VOCs profile in urine. There are limited data on the effect of bowel preparation on VOCs production in vivo. With further studies of VOCs in patients with symptoms of gastrointestinal disease, the effect of gut preparation on their abundance will need to be quantified. But before going further, there are still some issues that need to be addressed.

1The gap between the arrow and the box in Figure 1 is not consistent.

2I don't know if it is a typographical problem, but the figure notes in Figure 1 are not shown completely and the figure notes are not centered.

3Please standardize the abbreviations for Volatile Organic Compounds, some are VOC and some are VOCs in the paper.

4The spacing of the notes in the three tables is not uniform, and the notes in Table 2 are not centered.

5The rightmost part of Table 1 and Table 3 is not fully displayed, probably due to formatting.

6References are not aligned with other subheadings.

7The conclusion section can add some outlook on this research direction as appropriate.

8This work investigated nanomaterials enabled VOCs detection. Some relative papers may enrich the concepts and background of this work as references: ACS Appl. Mater. Interfaces, 2022, 14: 7301-7310; Sens. Actuators B: Chem, 2022, 370, 132441; Materials Today Physics, 2023, 30: 100951.

Author Response

1、The gap between the arrow and the box in Figure 1 is not consistent.

RESOLVED - Changed arrow distance to ensure consistency. 

2、I don't know if it is a typographical problem, but the figure notes in Figure 1 are not shown completely and the figure notes are not centered.

RESOLVED --> The text within the Figure has been centred and is now entirely shown.  

3、Please standardize the abbreviations for Volatile Organic Compounds, some are VOC and some are VOCs in the paper.

RESOLVED --> The paper has now been corrected so that the following only applies:

VOC used when singular i.e. volatile organic compound

VOCs now used when the plural applies i.e. volatile organic compounds

Previously there was incongruency between use of VOC and VOCs – now VOC is only used when abbreviating “volatile organic compound” and VOCs for “volatile organic compounds”.

4、The spacing of the notes in the three tables is not uniform, and the notes in Table 2 are not centered.

RESOLVED --> The notes/text in Table 2 are now centered. The spacing of the text in the three tables has now been made more equivalent and is as close to uniform as possible but due to the format of the required tables it is not possible to alter further without losing the rightmost part of the tables. 

5、The rightmost part of Table 1 and Table 3 is not fully displayed, probably due to formatting.

RESOLVED --> The rightmost parts of Table 1 and Table 3 are now fully displayed

6、References are not aligned with other subheadings.

RESOLVED --> This has now been appropriately adjusted

7、The conclusion section can add some outlook on this research direction as appropriate.

RESOLVED --> Thank you. We have now added further detail on the requirement of future research in this direction and the methods with which it should be undertaken

8、This work investigated nanomaterials enabled VOCs detection. Some relative papers may enrich the concepts and background of this work as references: ACS Appl. Mater. Interfaces, 2022, 14: 7301-7310; Sens. Actuators B: Chem, 2022, 370, 132441; Materials Today Physics, 2023, 30: 100951.

RESOLVED --> Thank you. We have read these papers and used them to enrich the introduction with one reference.

Many thanks for your useful comments and suggestions.